# Eye-Related COVID-19: A Bibliometric Analysis of the Scientific Production Indexed in Scopus

**DOI:** 10.3390/ijerph19169927

**Published:** 2022-08-11

**Authors:** Verónica García-Pascual, Elvira García-Beltrán, Begoña Domenech-Amigot

**Affiliations:** Department of Optics, Pharmacology and Anatomy, University of Alicante, 03690 Alicante, Spain

**Keywords:** COVID-19, Scopus, network analysis, bibliometrics, eye manifestations, eye

## Abstract

This paper analyzes, from a bibliometric viewpoint, those publications that relate COVID-19 and eye indexed in Scopus since the beginning of the pandemic, and it identifies the resulting main research lines. A bibliographic search in the Scopus database was conducted for publications that simultaneously include ocular and visual manifestations and aspects with COVID-19, from 1 January 2020 to 16 March 2021, and the obtained bibliographic information was processed with VOSviewer (v. 1.6.16). A total of 2206 documents were retrieved, and 60% were original articles. The USA published the most studies (24.6%). The retrieved documents had a total of 18,634 citations. The h index of the set of retrieved documents was 58. The *Indian Journal of Ophthalmology* was the most productive journal, while *JAMA Neurology* and *The Lancet* accounted for 20% of citations. Three keyword clusters representing hotspots in this field were identified. Eye-related COVID-19 research is an emerging field with plenty of scientific evidence whose growth is expected to increase as the aspects and manifestations of the disease, its treatment and the effect of vaccination on it become known.

## 1. Introduction

In December 2019, a group of pneumonia cases of unknown etiology were reported among some people who had been in a market in the city of Wuhan. A few days later, the Chinese Health Authorities identified and sequenced an outbreak of a new coronavirus called SARS-CoV-2 as the causal agent. Since then, and until now (20 July 2022), according to official figures, there have been more than 559 million contagions, and 6.3 million people have died from this disease worldwide [1]. The WHO considers that the number of people who have died directly or indirectly from this disease could be triple that official figure.

As the virus rapidly spread around the world, many researchers focused their activity on better understanding this disease in an attempt to control the pandemic. This led to an unparalleled amount of scientific literature about COVID-19 and the virus causing it [2,3,4,5,6,7]. Logically, the first studies were conducted in the fields of life sciences and health (genetics, clinical manifestations of the disease, concomitant diseases, etc.), but the theme rapidly became cross-sectional, in which many knowledge areas became involved working on different research lines [8,9,10,11,12,13]. Thorough reviews and bibliometric analyses have been performed to enable the search for these studies and to synthesize and facilitate their findings.

Since the disease emerged, different works have shown some eye-related COVID-19 findings, as well as implications of the pandemic for visual health professionals [13,14]. The most frequent themes reveal clinical eye symptoms of the disease, particularly inflammatory changes on the ocular surface [15], the role played by this surface and tears in transmitting the disease, conjunctival swabs, tear samples to clinically diagnose the disease [16], and retina findings [17,18], of which retinal vascular occlusions or bilateral optical coherence tomography (OCT) anomalies are highlighted. Another relevant research line is related to personal protective equipment (PPE) and implications in visual health professionals offering their services [19]. Strategies to control risk of infection in clinical practice by taking administrative and environmental control measures and the use of PPE have been studied [20]. One of the high-risk activities is adapting contact lenses and administering their use and a considerable number of studies have dealt with this matter [21,22,23]. Finally, some works have been published about the impact of COVID and lockdown on the typical academic activities of teaching optometry [24] and contactology training [25]. Recently, two bibliometric analysis focused on developments in COVID-19 research in the ophthalmology field have been published [26,27]. However, we found that some of the issues above mentioned issues lay beyond the field of ophthalmology, ranging from occupational risk prevention to academic aspects which are beyond the search strategy posed in these two papers.

The aim of this study was to perform a bibliometric analysis of publications related to COVID-19 and the eye that were indexed in Scopus in the first 14 months of the pandemic, and to identify the main research lines that have emerged as a result of this crisis. This study was set out as an exhaustive analysis of both ocular findings and manifestations and any other relevant aspects in the field of care and prevention of the visual and vision system.

## 2. Materials and Methods

### 2.1. Data Search Strategy

To conduct this study, the Scopus database was used (Elsevier BV Company, Amsterdam, The Netherlands), which is one of the largest traditional bibliographic databases to have been validated in several bibliometric analyses [28,29].

Search keywords included: “eye”, “vision”, “visual acuity”, “ocular disease”, “ocular symptoms”, “ocular implications” “conjunctivitis”, “keratoconjunctivitis”, “tears”, “cornea”, “optometr*”, “ocular findings”, “ocular manifestations”, “ophthalmic manifestations”, “ocular surface”, “conjunctival secretions”, “conjunctival swabs”, “ocular signs”, “conjunctiva” “covid-19”, “covid 19”, “covid19”, “CoronaVirus”, “corona virus”, “coronavirus disease”, “SARS-CoV-2”, “2019-nCoV”, and “coronaviruses” in the Article title, Abstract and Keywords fields. The search was limited to documents published in any language from 1 January 2020 to 16 March 2021.

Books, book chapters, errata documents, corrections and conference proceedings were excluded because such proceedings include abstracts that can be published in proceedings books and as full papers in journals, which could lead to false positives.

### 2.2. Data Analysis and Visualization

The obtained Scopus records were exported as csv (comma-separated values file) to then be imported to the VOSviewer networks visualization software. The exported data included the h-index of the set of documents (a scientist has index h if his or her Np paper has at least h citations each and the other (Np-h) papers have ≤h citations each), document type, authors, institutions, countries, sources, titles, keywords, total citations (including self-citations) and the bibliographic references of each register to create science maps [30].

The bibliographic information obtained during the search was processed with the VOSviewer (v. 1.6.16) software developed at the Centre for Science and Technology Studies (CWTS) of Leiden University to create and visualize bibliometric maps [31]. The maps obtained with this software are maps based on distance (the shorter the distance, the closer their relation), which enables a graphic visualization of elements to be obtained by labelled nodes. Bibliographic records are objects of interest, and each element in the register (source, authors, country, etc.) is represented by a node. Node size varies according to its weight or importance: the bigger the node, the more relevant the element (highly cited publications, highly prolific researchers, etc.). Nodes are positioned in a two-dimensional space in such a way that strongly related nodes are located close to each other while weakly related nodes are located far away from each other. This facilitates grouping related elements (items). Together, both elements and links form networks grouped into clusters [30]. Networks may consist of several thousand nodes. Specifically, networks of co-authorship of countries, co-citation of scientific journals, citation and co-citation of authors and co-occurrence of keywords and indexing terms have been obtained. Two publications are co-cited if there is a third publication that cites both publications. The larger the number of publications by which two publications are co-cited, the stronger the co-citation relation between the two publications. Regarding the co-occurrence relations, the number of co-occurrences of two keywords is the number of publications in which both keywords occur together in the keyword list. Different thesauri were prepared to disambiguate the various forms that researcher names and sources come in, to combine synonymous and quasi-synonymous terms, to correct spelling differences and to remove irrelevant terms, which were added while creating maps.

The data used in this study came from an open database and did not involve human subjects. Therefore, the requirement for approval by the institutional review committee is exempt.

## 3. Results

### 3.1. Analysis of Document Output

In all, 2206 documents were retrieved, and most were original research papers. The mean of monthly articles was 145 in 2020 but was 185 in 2021 when only 2.5 months were evaluated. Table 1 offers the distribution of documents according to their typology. Most retrieved documents were published in English (93.70%). All the documents received contained 18,634 citations in total. The h-index of the set of retrieved documents was 58.

Eye-related COVID-19 is a multidisciplinary theme, as Table 2 shows, because although the health sciences and medicine fields predominate, studies have also been found in fields such as social sciences, engineering or computer sciences.

### 3.2. Analysis of Major Countries/Regions

Researchers from more than 200 countries have contributed to publish the retrieved documents on eye-related COVID-19. Figure 1 shows the geographical distribution of their publications on a world map. The most productive country is the USA with 24.6% of documents, followed by India with 12.5% and the UK with 11.4%.

The visualization of co-authorship among the countries with a minimum productivity of 15 documents appears in Figure 2. This map shows that 34 countries are distributed in six different clusters. A cluster is a set of closely related countries. Red and purple are European clusters, green is Asiatic cluster, and blue and yellow are American clusters. The most relevant collaboration was between the USA and the UK (link strength = 39) and between the USA and China (link strength = 35).

### 3.3. Analysis of Production, Citations and Co-Citations of Journals

The 10 most productive journals are shown in Table 3. The *Indian Journal of Ophthalmology* and *Graefe’s Archive for Clinical and Experimental Ophthalmology* were the most productive journals, and each one had practically (2%) all the citations. They all belong to the ophthalmology category, except for the *Journal Medical Virology*, which is classified in infectious diseases. The journal with the highest h-index is *Ophthalmology*. The source of the indicators and subject categories is the Scimago Journal & Country Rank.

Figure 3 shows a co-citation map of journals with a minimum of 150 citations. Of the 20,114 cited sources, only 35 met the criterion, and three clusters in all were obtained. The journals belonging to the same group were those that tended to be jointly cited. They are represented in the same color. Red mainly denotes general medicine and is the biggest group formed by 16 journals, where most citations concentrate. Of those with the most citations in this first cluster, we find *The Lancet*, *New England Journal of Medical*, *Jama-Journal of the American Medical Association* and *Journal of Medical Virology*. Green denotes the predominant ophthalmology journals. This group is made up of 11 journals, and the most co-cited one is *Ophthalmology* with 633 citations. Finally, red mostly depicts multidisciplinary journals, such as *Nature* and *PLoS ONE* with 569 and 448 received citations, respectively. 

The close co-citation relations that appeared between journals were *The Lancet–New England Journal of Medicine* (link strength = 3561) and *New England Journal of Medicine–JAMA—Journal of the American Medical Association* (link strength = 1922). The three groups that form the map are quite close, but the distance between the ophthalmology group (the green cluster) and the general medicine group (the red cluster) is longer. This last group occupies the most central position, which indicates a relation with the other two groups. It is worth stressing that some of the most productive journals in our search do not appear on the co-citations map. The total number of citations received for the set of retrieved documents is 18,634. Figure 4 shows the 10 journals with the most citations, where we can see that only two journals, *JAMA Neurology* and *The Lancet*, obtained almost 20% of the citations.

### 3.4. Analysis of Authorship Pattern and Collaboration

In all, 10,489 researchers participated in publishing the retrieved documents with a mean of 4.8 authors per document. Collaborative research prevailed with 86%. The most productive author was Sharma, Namrata K with 16 publications, followed by Agrawal, Rupesh V and Shetty, Rohit with nine publications each.

Figure 5 is the map of the citations of the authors with a minimum productivity of five documents and a minimum number of 50 received citations. The map includes 56 nodes (authors) and seven clusters. The closest nodes indicate those authors who closely collaborate in research work.

Further information that can be analyzed from our results is who has been cited in the works retrieved in our search from a list of references of each document. Figure 6 is the map of authors’ co-citations for which the authors with a minimum of 55 citations were chosen. We see that 184 authors of a total of 83,315 meet this criterion. The map also clearly shows the three created clusters; the red cluster contains the most authors with 89. Although many authors coincide on both maps (citation and co-citation), we see that the co-citation map contains many more nodes because they were obtained from different register fields. The authors (node) on the citation map were obtained from the register’s authors field, where the nodes in co-citation were obtained by emptying the list of references for each register.

### 3.5. Analysis of Highly Cited Documents

Table 4 shows the 10 most cited documents in the eye-related COVID-19 literature, of which seven are research papers, two are letters and one is a review. The most visible and influential article (1741 citations) is about the neurological manifestations of those patients hospitalized in Wuhan, which points out visual disorders among peripheral nerve system manifestations. Three of the articles on this list (6, 8 and 9) are specifically about COVID-19 and ocular manifestations. Study 4 describes a series of pediatric cases, which was retrieved because “conjunctivitis” is an indexed term in it for being a frequent manifestation in the described case series. We also find one review published in *The Lancet* about measures to prevent COVID-19 spreading, including eye protection (glasses, face masks, etc.). All the others generally do not present such a close relation to ocular and visual manifestations, and were the main objective of this search. For example, Study 2 studies the expression of the genes associated with the virus entering different tissues, including ocular tissues. Study 5 presents a series of prevention and control measures taken during dental practice to avoid contagion from eye mucus. Finally, two clinical assays have been carried out with hydroxychloroquine. In one “blurred vision” appears as an indexed term, and lack of eye protection appears in the other, and is one of the groups that helps to differentiate the degree of being exposed to the virus.

When ordering the retrieved documents according to the relevance conferred by Scopus and by focusing on the 10 most relevant ones (Table 5), we found only one article in common with the list shown in Table 4. The most relevant documents included seven research papers and two reviews. The article considered to the most relevant one was “*The Ocular Manifestations and Transmission of COVID-19: Recommendations for Prevention*”, published in the *Journal of Emergency Medicine* in 2020 with 18 received citations. All the documents classified as relevant were specific of COVID-19 and ocular manifestations, signs or treatments. Seven of them were published in the journals belonging to the ophthalmology category.

### 3.6. Analysis of Co-Occurrence of Terms

Figure 7 shows a co-occurrence map where the chosen analysis unit was authors’ Keywords. The term COVID-19 appears as the most important node with 890 occurrences. The greatest link strength appeared between COVID-19-SARS-COV-2 (link strength = 247) and was 50 and 47, respectively, for COVID-19 conjunctivitis and COVID-19 ophthalmology.

Figure 8, which contains the co-occurrence map of terms in which the analysis units were both authors’ keywords and indexation terms. The latter belong to controlled languages, and more accurately represent the content of documents. The different colors on the map allow the three clusters to be analyzed in which the program’s algorithm classified the different terms.

The term COVID-19, which covers the search in our field, was the biggest and most centered node. This indicates a marked co-occurrence with other nodes. The most occurrent terms were pandemic, coronavirus infection, pneumonia, viral, among others.

From the clusters or groups, different topics of interest are seen in which the obtained information can be classified. Table 6 shows the most frequent terms in all the groups, along with a proposal of the cluster name according to the content of documents.

## 4. Discussion

This study is the first exhaustive bibliometric analysis of the scientific production on COVID-19 and ocular findings, manifestations and any other relevant aspect in the care and prevention field of the visual and vision system indexed in Scopus. During the short study period (from the start of the pandemic to 16 March 2021), original research papers were mainly retrieved in English and in different theme categories. However, a very high percentage of retrieved registers (18%) were letters and editorial material, probably due to the pandemic’s socio-scientific impact. As expected, in 2021, production grew more than in 2020. The search was repeated in July 2022, and the monthly rate of articles rose from 185 in 2021 to 314 in 2022.

Scientific production in this field is dominated by the USA, along with India, China and the UK. This is not a surprising result because these countries, with the exception of India, head the world’s scientific production ranking both in the set of all disciplines and in the medicine category of Scimago Journal & Country Rank, for the 1996–2020 period. India occupies a lower position in this ranking (7th in the set of all disciplines and 11th in medicine). Collaboration among countries is headed by the USA, and the most well-established co-authorship is that of the USA and the UK.

The map of author citations shows differences between an author’s production and influence; for example, the most productive author is barely seen on the citations map. As the authors’ co-citation map was obtained from the bibliographic references from our results, mainly researchers in Asian countries appear on this map because, logically, Asian countries were the initial leaders of scientific productivity. We can also see that the most productive journals are neither those receiving the most citations nor the most influential ones. For instance, the most productive journal, according to our results, is *Indian Journal of Ophthalmology*, but it does not appear on the map of journals’ co-citations. This might indicate a significant isolation index. Nor does it appear among those receiving the most citations. It must be taken into account that it is a journal that occupies an average position in a specific field, ophthalmology, whose median number of citations cannot be compared with other general journals such as *The Lancet* or *New England of Medicine* or with journals from other fields, such as *JAMA Neurology*. This suggests differences in production (productivity indicators) and consumption (visibility and impact indicators).

A keywords and indexing terms co-occurrence analysis was carried out using the VOSviewer software to visualize and identify the main theme domains for research into eye-related COVID-19, and the interest of scientists and publishers. After analyzing terms, a decision was made to group terms into three large clusters. To this end, providing a name was proposed according to the theme of the articles that contain and represent current topics of interest.

The top 10 most cited publications contain some documents in which the eye subject was not the main subject. For example, the first study in this ranking describes all the neurological signs noted in the hospitalized patients in Wuhan, which indicates that visual disorders were displayed as peripheral nerve system manifestations. This was an expected result because of the search strategy using natural language terms whose main characteristic was a lack of being univocal. For instance, one of the most ambiguous terms in our equation was the term eye. Although this search can produce certain documental noise, all the documents were about the eye theme to a greater or lesser extent. In an attempt to obtain a relation of the most significant documents, we performed a second ranking based on the Scopus relevance algorithm. In this case, the results were more accurate, and only one document was common on both lists, mainly because this algorithm considers if terms appear in a work’s title and in authors’ keywords. This is why the works on this list match our search objective better. The visibility and impact of most journals on this last list can be considered acceptable because they occupy medium and high positions in the SJR. This result must be interpreted somewhat cautiously given the interest and structures implemented by publishing platforms to prioritize publishing COVID-19-related works in an unparalleled effort to communicate findings as soon as possible. Indeed some experts have pointed out that this effort has somewhat decreased the quality of works and slowed down the publication of other research themes [51].

Since the pandemic began, many bibliometric analyses have been published with the term COVID-19. Some of them have evaluated overall scientific production, and others, such as our study, have related it to another scientific field. Among the latter, we find some similar to our own, such as COVID-19 and university learning aspects [52], effects on the socio-scientific research panorama [53], COVID-19 and diabetes [54], effect of COVID-19 on the natural environment [55], applying artificial intelligence during the COVID-19 pandemic [56], the growth of the medical literature on COVID-19 using evidence maps [57], COVID-19 articles published in odontology journals [12], have followed similar methodologies to the one that we followed. Recently, two scienciometric analysis of COVID-19 and ophthalmology were published. They employed the Web of Science database and limited the search to ophthalmology [53] in the science category field. Logically, its results were more limited (616 research items vs. the 2206 that we retrieved in our study), firstly because it resorted to a database with less coverage, and secondly it did not retrieve generalist works which also describe visual/ocular problems, or problems related to eye protection or contact lenses. This was because we wanted to make our search as thorough as possible.

## 5. Limitations and Strengths

Like other bibliometric studies [26,28,29], the present study has a few limitations inherent to the database and the employed equation. As previously mentioned, some retrieved documents do not deal with ocular aspects as the work’s major topic because it is not possible to examine Scopus using documental language. Another limitation is having employed only one database. Despite Scopus being one of the largest databases, this analysis did not identify all the published materials (grey literature, publications in non-indexed journals, repositories, etc.). Another of its limitations is that assigning the country to which each document is attributed is performed with author affiliations, which favors international collaborations, because one same work is assigned to different countries, thus adding one article unit to each country, unlike publications by teams with the same affiliation [55], which only count as one unit. Finally, as with any bibliometric study, the presented results reflect the tendencies observed on the date when the search was conducted, and the results are exported to be analyzed. By way of example, the NCit value in Table 3 and Table 4 is currently higher, and the ordering in them all may have been modified. Moreover, as our search was conducted in a very short time window (just under 15 months), it would be worthwhile for future studies to complete this search over a longer period to verify to what extent these tendencies remain.

The strong point of this work lies in it being the first study to thoroughly analyze from a bibliometric viewpoint works about COVID-19 and eyes that have been published in scientific journals indexed in Scopus between January 2020 and March 2021. Apart from studying authors, countries and sources, this study allowed the visualization of three keyword clusters that represent hotspots in the eye-related COVID-19 field.

The main message of this research is that it shows the impact of COVID-19 not only in the area of general medicine and ophthalmology, but also in other areas such as personal protective equipment (PPE), implications in visual-health professionals, the high-risk adaptation and use of contact lenses or the impact of lock-down on the usual academic activities involved in teaching visual sciences. Future studies should analyze other emerging themes in the eye–COVID-19 relationship, such as the impact on sight-related aspects of vaccines or of long-term COVID. This can be achieved by implementing the same search strategy with different time limits to observe new trends and themes that may become obsolete.

## 6. Conclusions

To conclude, this bibliometric analysis revealed that eye-related COVID-19 research is an emerging field with scientific evidence whose growth is expected to increase as the aspects and manifestations of the disease, its treatment and the effect of vaccination become known.

## Figures and Tables

**Figure 1 ijerph-19-09927-f001:**
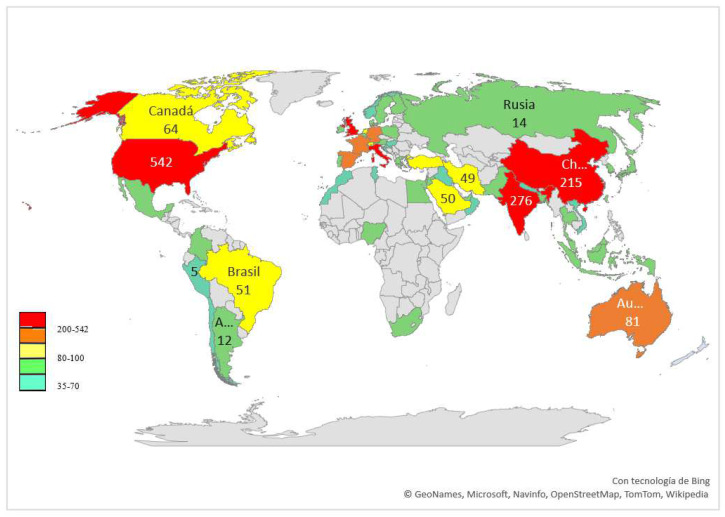
Geographical distribution of publications on eye-related COVID-19. Map color is coded where the world regions in red had the highest productivity, and those in green and blue the lowest productivity.

**Figure 2 ijerph-19-09927-f002:**
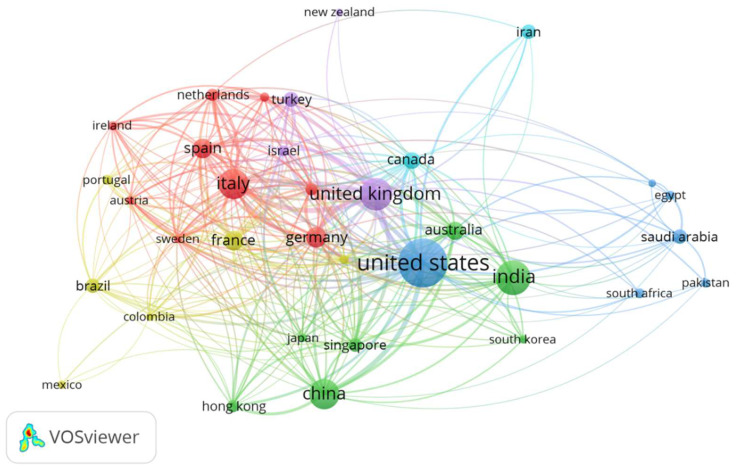
Network visualization map of international collaboration among countries with a minimum productivity of 15 documents. The thickness of the connecting line between two countries shows the strength of collaboration. Countries with a similar color form one cluster. For example, the countries in red, such as Germany, Spain and Italy, are found in one cluster with the highest collaboration in this cluster.

**Figure 3 ijerph-19-09927-f003:**
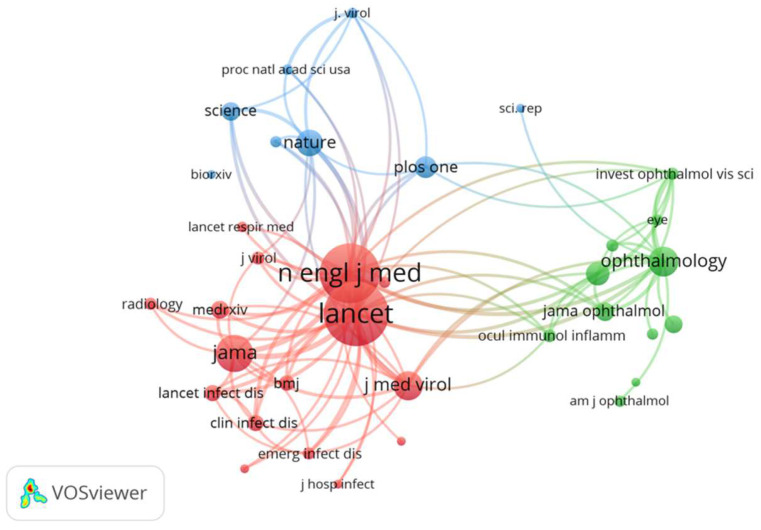
Network visualization map of the co-citation analysis for the journals with published documents about eye-related COVID-19 with a minimum of 150 citations. Journals with the same color are often co-cited.

**Figure 4 ijerph-19-09927-f004:**
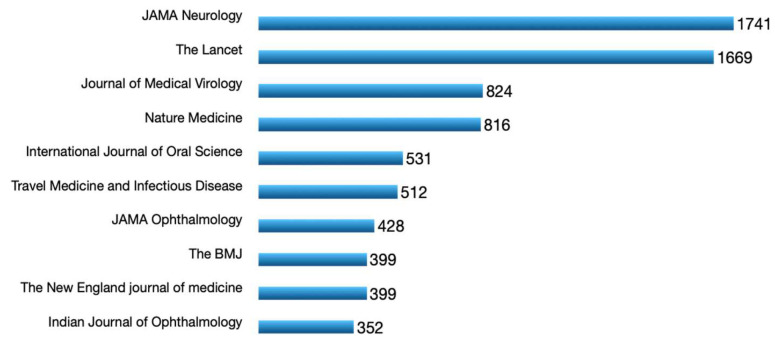
Top 10 cited sources of eye-related COVID-19 literature.

**Figure 5 ijerph-19-09927-f005:**
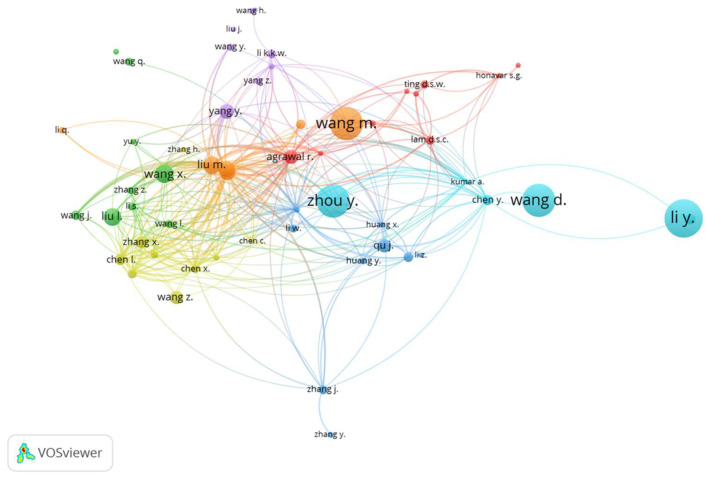
Network visualization map of the author citation analysis. Highly cited authors are indicated by the larger size of the node. Y. Li is the author with the highest number of citations (2351). Blue and orange are neurology clusters, while in the red cluster ophthalmology predominates and in purple we represent authors from fields such as computer, biostatistics, and public health.

**Figure 6 ijerph-19-09927-f006:**
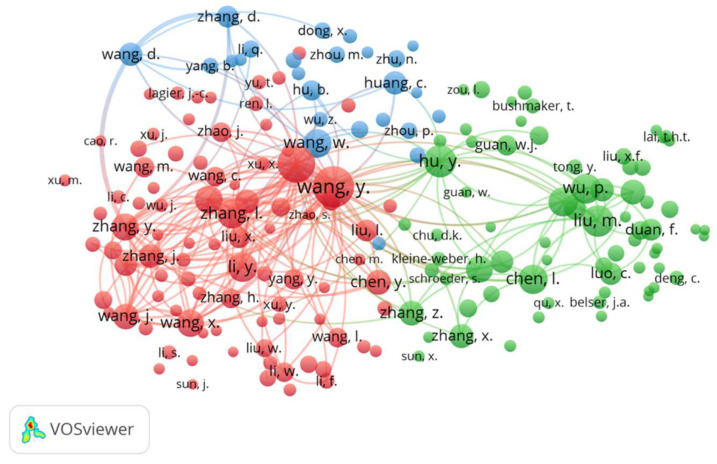
Network visualization map of the journal co-citation analysis for the authors with published documents on COVID-19 and eyes with a minimum of 55 citations. The authors with the same color are often co-cited. The most co-cited authors of the green cluster are specialized in virus transmission and ocular findings related to COVID-19, in the field of ophthalmology. The most co-cited authors of the red cluster work in clinical features, infectious diseases, and radiological findings, mostly in the respiratory medicine field. Blue cluster represents the authors who worked on virus identification using molecular techniques and genetic sequencing.

**Figure 7 ijerph-19-09927-f007:**
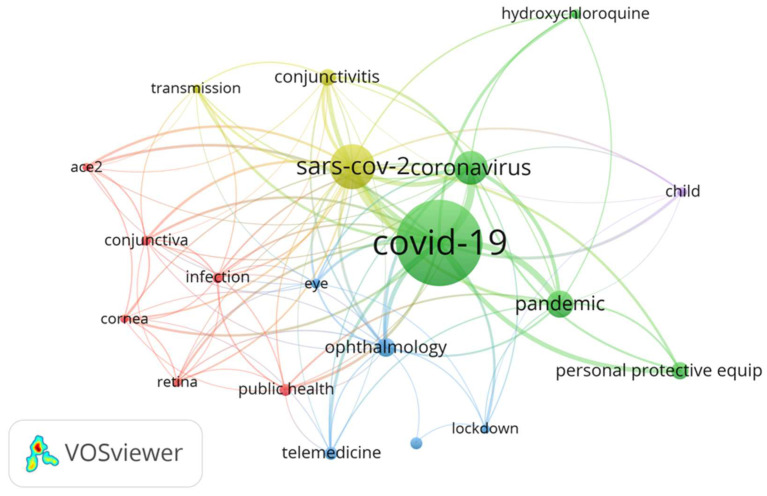
Network visualization of the co-occurrences relations between authors’ keywords. The keywords with minimum occurrences (20) are shown on the map. Label size is an indication of the frequency of each keyword’s occurrence, while different colors represent word clusters. The keywords of the same cluster are often listed together. For example: conjunctiva, infection and public health are shown with the same color because they are closely related and usually co-occur.

**Figure 8 ijerph-19-09927-f008:**
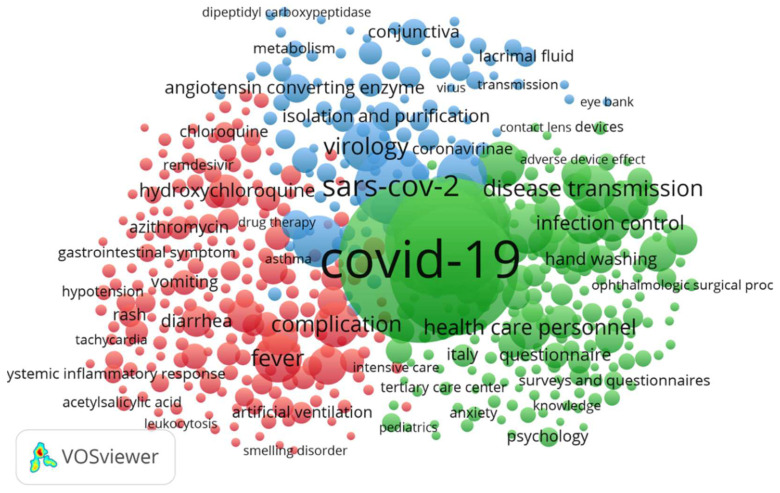
Network visualization of the co-occurrences relations between all the keywords (authors’ and index keywords). Keywords with minimum occurrences (15 times) are shown on the map. Label size is an indication of the frequency of each keyword’s, and different colors represent word clusters. The words with the same cluster are often listed together.

**Table 1 ijerph-19-09927-t001:** Types of retrieved documents (2019–2020).

Document Type	*n*	% N = 2206
Article	1326	60.11
Review	356	16.14
Letter	271	12.28
Note	116	5.26
Editorial material	115	5.21
Short survey	11	0.50
Conference paper	10	0.45
Data article	1	0.05
Total	2206	100

**Table 2 ijerph-19-09927-t002:** Classification according to thematic area.

	% N = 2206
Medicine	54
Social sciences	8
Neuroscience	7
Biochemistry, genetics, and molecular biology	5
Pharmacology, toxicology, and pharmaceutics	4
Immunology and microbiology	4
Computer science	3
Health professions	3
Environmental science Engineering others	3 6

**Table 3 ijerph-19-09927-t003:** Top 10 active journals in publishing in eye-related COVID-19 literature (2019–2021) N = 2206.

	Source	H-Index	Country	Rank	Citations	Frequency	%
1	Indian Journal of Ophthalmology	47	India	Q3	352	108	4.90
2	Graefe’s Archive for Clinical and Experimental Ophthalmology	96	Germany	Q1	346	33	1.50
3	Eye (Basingstoke)	93	UK	Q1	274	30	1.36
4	European Journal of Ophthalmology	51	Italy	Q2	40	28	1.27
5	Ocular Immunology and Inflammation	53	UK	Q2	303	23	1.04
6	JAMA Ophthalmology	190	USA	Q1	428	22	1.00
7	Journal of Medical Virology	111	USA	Q2	824	21	0.95
8	Ophthalmology	229	Netherlands	Q1	300	21	0.95
9	Journal Francais d’Ophtalmologie	29	France	Q3	66	20	0.91
10	Journal of Community Eye Health	20	UK	Q4	13	18	0.82

Source H-Index: Scopus Scimago; source rank: Journal & Country Rank; citations received about our search results.

**Table 4 ijerph-19-09927-t004:** Top 10 cited articles in the eye-related COVID-19 literature (2019–2021). Data retrieved in 16 March 2021.

Ranking	Document Title	Authors, Year of Publication	Journal	NCit	Document Type
1	Neurologic Manifestations of Hospitalized Patients with Coronavirus Disease 2019 in Wuhan, China [32]	Mao L., Jin H., Wang M. et al., 2020	*JAMA Neurology*	1741	Article
2	SARS-CoV-2 entry factors are highly expressed in nasal epithelial cells together with innate immune genes [33]	Sungnak W., Huang N., Becavin C. et al., 2020	*Nature Medicine*	572	Article
3	Physical distancing, face masks, and eye protection to prevent person-to-person transmission of SARS-CoV-2 and COVID-19: a systematic review and meta-analysis [34]	Chu D.K., Akl E.A., Duda S. et al., 2020	*The Lancet*	572	Article
4	Hyperinflammatory shock in children during COVID-19 pandemic [35]	Riphagen S., Gomez X., Gonzalez-Martinez C. et al., 2020	*The Lancet*	555	Letter
5	Transmission routes of 2019-nCoV and controls in dental practice [36]	Peng X., Xu X., Li Y. et al., 2020	*International Journal of Oral Science*	531	Review
6	Evaluation of coronavirus in tears and conjunctival secretions of patients with SARS-CoV-2 infection [37]	Xia J., Tong J., Liu M. et al., 2020	*Journal of Medical Virology*	426	Article
7	A Randomized Trial of Hydroxychloroquine as Postexposure Prophylaxis for Covid-19 [38]	Boulware D.R., Pullen M.F., Bangdiwala A.S. et al., 2020	*The New England Journal of Medicine*	399	Article
8	2019-nCoV transmission through the ocular surface must not be ignored [39]	Lu C.-W., Liu X.-F., Jia Z.-F., 2020	*The Lancet*	395	Letter
9	Characteristics of Ocular Findings of Patients with Coronavirus Disease 2019 (COVID-19) in Hubei Province, China [40]	Wu P., Duan F., Luo C. et al., 2020	*JAMA Ophthalmology*	354	Article
10	Hydroxychloroquine in patients with mainly mild to moderate coronavirus disease 2019: Open label, randomised controlled trial [41]	Tang W., Cao Z., Han M. et al., 2020	*The BMJ*	347	Article

NCit: Citations received about the search results.

**Table 5 ijerph-19-09927-t005:** Top 10 most relevant articles (Scopus) in the eye-related COVID-19 literature (2019–2021). Data retrieved in 16 March 2021.

Ranking	Document Title	Authors, Year of Publication	Journal	NCit	Document Type
1	The Ocular Manifestations and Transmission of COVID-19: Recommendations for Prevention [42]	Dockery D.M., Rowe S.G., Murphy M.A. et al., 2020	*Journal of Emergency Medicine*	18	Article
2	Detection of SARS-CoV-2 in conjunctival secretions from patients without ocular symptoms [43]	Li X., Chan J.F.-W., Li K.K.-W. et al., 2020	*Infection*	4	Article
3	COVID-19 and the Ocular Surface: A Review of Transmission and Manifestations [44]	Ho D., Low R., Tong L. et al., 2020	*Ocular Immunology and Inflammation*	16	Review
4	Ocular Features and Associated Systemic Findings in SARS-CoV-2 Infection [45]	Cavalleri M., Brambati M., Starace V. et al., 2020	*Ocular Immunology and Inflammation*	4	Article
5	The role of the ocular tissue in sars-cov-2 transmission [46]	Peng M., Dai J., Sugali C.K. et al., 2020	*Clinical Ophthalmology*	2	Review
6	Characteristics of Ocular Findings of Patients with Coronavirus Disease 2019 (COVID-19) in Hubei Province, China [40]	Wu P., Duan F., Luo C. et al., 2020	*JAMA Ophthalmology*	354	Article
7	Update and Recommendations for Ocular Manifestations of COVID-19 in Adults and Children: A Narrative Review [47]	Danthuluri V., Grant M.B., 2020	*Ophthalmology and Therapy*	3	Review
8	How to approach management of ocular surface disease during COVID-19 pandemic? [48]	Labetoulle M., Doan S., Rousseau A., 2020	*Journal Francais d’Ophtalmologie*	-	Article
9	The prevalence of conjunctivitis in patients with novel coronavirus (COVID-19) and preventive measures [49]	Gazizova I.R., Desheva Y.A., Gavrilova T.V. et al., 2020	*Russian Journal of Clinical Ophthalmology*	1	Article
10	Transmission of COVID-19 through eyes-review article [50]	Ingole S.S., Bhutada R., 2020	*International Journal of Research in Pharmaceutical Sciences*	-	Article

NCit: Citations received about the search results.

**Table 6 ijerph-19-09927-t006:** Proposal of names for the clusters formed by VOSviewer based on their degree of relatedness. Word cluster colors match those employed in Figure 8.

Nodes/Representative Items	Proposed Cluster Name
Hydroxychloroquine, Cloroquine, C reactive protein, Azithromycin, Pneumonia, Blurred vision, Gastrointestinal symptom, Headache, Fever, Myalgia, Cough, Dyspnea, diarrhoea, Disease severity, Diagnostic imaging, Complication, polymerase chain reaction, Mortality	(Red cluster) Symptoms and signs,
COVID–19, Virus pneumonia, Ophthalmology, Pneumonia viral, Coronavirus infection, Eye disease, Emergency health service, Patient care, Health care personnel, Telemedicine, Hand washing, Ophthalmologist, Disease transmission, Infection prevention, Infection control, Virus transmission, Procedures, Infection risk, Epidemiology, epidemic, Personal protective equipment, Eye protection, Ophthalmologist organisation and management, pandemic, patient care	(Green cluster) Exposure risks, protection, epidemiology
Genetics, Sars-cov-2, Sars virus, Sars coronavirus Coronavirus, Virology, Conjunctiva, Cornea, Tears, Lacrimal fluids, Eye, Angiotensin converting enzyme, Conjunctivitis, Viral conjunctivitis, Pathology, virus rna	(Blue cluster) Clinical pathology, etiology and causative organism

## Data Availability

Not applicable.

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
