# Peer review of "Eye-Related COVID-19: A Bibliometric Analysis of the Scientific Production Indexed in Scopus"

_ijerph, 2022, doi:10.3390/ijerph19169927_

Round 1

Reviewer 1 Report

IJERPH-1805130

Eye-Related COVID-19: A Bibliometric Analysis of the Scientific Production Indexed in Scopus

This is a well-written paper on an interesting topic. The authors do have knowledge of bibliometric analysis but they are probably not experts in the advanced bibliometric methodology. Although this is not very necessary for a public health journal and for the predominantly medical readers, I think several improvements can be made, and that is the focus of my review. I structure my review according to the sections of the manuscript.

Section 2.2: are citations corrected for self-citations? If not, please indicate this in the text.

Section 3.1: The distribution of citations over publications is very skew. So statistically the mean is not a very appropriate measure. It would be more interesting for the reader to present the citation distribution.

h-index: It is not clear in advance for the reader what 58 means, is it high or low? This number should be given a context, for instance by comparing it with the other articles in the analyzed journals. Otherwise it gives hardly any meaningful information.

Fig.3: Of course the colors indicate some form of collaboration with an intensity above a specific threshold determined by the VOS viewer algorithms, but that is the case for all colors, so it would be informative to give more explanation on the various colors, for instance red is a European cluster, green is typically Asian, etc.

Section 3.3: The readers of a public health journal, such as the one to which this paper is submitted, do not necessarily have some bibliometric knowledge. Therefore the authors should briefly explain, perhaps in a text-box, what co-citation is, and what interesting information this method has to offer. Particularly that 'jointly cited' is not at the level of individual publications (as is mostly the case in co-citation analysis) but that we here deal with co-citation at level of entire journals. The consequence then, however, is that unexpected and rare, but highly interesting co-citations (for instance between an ophthalmology journal and an optical physics journal) disappear in the noise of the mass of the more common journal co-citations. Maybe the authors could do a preliminary analysis of such unexpected co-citations on the individual publication level.

Fig. 5: Does the ranking given in Fig. 5 based on eye-related COVID-19 publications differ substantially (or not) from the ranking of the same journals but based on the number of citations to all their publications? This would give interesting additional information on which journal has a high relative (instead of absolute) number of eye-related COVID-19 publications.

Fig. 6: Also here, explain the clusters/colors.

Fig. 7: Again, more explanation is necessary, first the co-citation at the level of journals was discussed, and here it is at the level of individual authors, but (in both cases) the basis of the method are the reference lists of individual publications. An important point here is also that co-citation structures reflect activities in the past, and quite often co-cited items (authors or publications) differ  substantially in age, for instance a paper (and thus its authors) from 1963 can be co-cited with a paper from 2020. Perhaps a figure would help.

Clearly the authors explored the possibilities of the VOS viewer. Why did they not make the bibliographic coupling analysis. The interesting aspect of bibliographic coupling (BC) is that it makes clusters of publications (or journals, or authors) on the basis of references these publications have in common. As it were the mirror of co-citation analysis with the advantage that the clusters are not ‘structures in the past’ (as is the case in co-citation analysis) but that BC clusters reflect structures of the current literature.

Furthermore, again a first explanation of the co-citation clusters/colors is necessary. There are many Chinese researchers, but some belong to the red and others to the green cluster. The only way to find out why researchers cluster red or green is to analyze the keywords the publications of, for instance, the most (co-)cited authors, for instance Y. Wang in the red cluster and L. Chen in the green cluster. Most probably the clusters indicate different specializations within eye-related COVID-19 publications, maybe different instrumentation, different epidemiological methods, things like that. That would give further relevant information for the reader, similar to what is done in the context of Figs. 8 and 9.

Section 3.5, Figs.8 and 9: Also here explanation is necessary to make the reader clear what a co-occurrence map is and what kind of information one can expect from such maps. I understand from the manuscript that the authors took the author-given keywords (Fig.8) as well as the combination of the author-given and index-given keywords (Fig.9). This type of keywords is rather conservative, mostly the more common terms. This has the advantage that the structures in the map are reasonably to understand. However, the VOS viewer also offers the possibility to extract (parse) terms from the titles and abstracts of the publications and this (co-occurrence based on text data) may give more unexpected, non-trivial keywords. The explanation of such text based maps however can be quite cumbersome.

Section 4: “We can also see that the most productive journals are neither those receiving the most citations nor the most influential. For instance, the most productive journal according to our results is Indian Journal of Ophthalmology, but it does not appear on the map of journals’ co-citations. This might indicate a significant isolation index. Nor does it appear among those receiving the most citations. This suggests differences in production (productivity indicators) and consumption (visibility and impact indicators)”.

Indeed, probably the publications in the Indian Journal of Ophthalmology are not very much cited, it is therefore important to find out the citation distributions per journal. Instead of an 'isolation index' (which is a quite interesting idea in itself..) I think it is more the low citation level of a journal that causes an 'isolation'. This could easily be analyzed and included in this manuscript.

Finally, I think that for the readers who are further interested in bibliometrics, references to informative overview articles could be helpful. Now there are hardly references to the basics of advanced bibliometric analysis. A very useful and comprehensive example is:  van Raan, AFJ 2019, ‘Measuring Science: Basic Principles and Application of Advanced Bibliometrics’, in W Glänzel, HF Moed, U Schmoch & M Thelwall (eds.), Handbook of Science and Technology Indicators, Series: Springer Handbooks, Springer, Heidelberg.

Author Response

see atachment

Reviewer 2 Report

Peer Review for International Journal of Environmental Research and Public Health (publisher MDPI)  

2022-07-12 

 Manuscript: García-Pascual, V.; García-Beltrán, E.; Domenech-Amigot, B. “Eye-Related COVID-19: A Bibliometric Analysis of the Scientific Production Indexed in Scopus” submitted to Int. J. Environ. Res. Public Health. 

General comment:

This manuscript is a bibliometric description of published articles on eye disease in covid. This topic is new and the results in this manuscript should be of interest to other researchers. 

This peer-reviewer referee, using pubmed search phrase “bibliometric on eye disease in covid” found a total of 4 articles:

1.    A Systematic Literature Review and Bibliometric Analysis of Ophthalmology and COVID-19 Research. Forouhari A, Mansouri V, Safi S, Ahmadieh H, Ghaffari Jolfayi A. J Ophthalmol. 2022 May 24;2022:8195228. doi: 10.1155/2022/8195228. eCollection 2022. PMID: 35646394 [This article was not found in the References Section] 

2.    Publication trends in telemedicine research originating from Canada. Xie JS, Nanji K, Khan M, Khalid MF, Garg SJ, Thabane L, Sivaprasad S, Chaudhary V. Healthc Manage Forum. 2022 May;35(3):153-160. doi: 10.1177/08404704211070240. Epub 2022 Apr 6. PMID: 35083937 [This article was not found in the References Section]

3.    COVID-19 and ophthalmology: A scientometric analysis. Kalra G, Kaur R, Ichhpujani P, Chahal R, Kumar S. Indian J Ophthalmol. 2021 May;69(5):1234-1240. doi: 10.4103/ijo.IJO_3284_20. PMID: 33913867 [This article was found in the References Section as Reference 53]

4.    Coronavirus disease 2019 (COVID-19): an evidence map of medical literature. Liu N, Chee ML, Niu C, Pek PP, Siddiqui FJ, Ansah JP, Matchar DB, Lam SSW, Abdullah HR, Chan A, Malhotra R, Graves N, Koh MS, Yoon S, Ho AFW, Ting DSW, Low JGH, Ong MEH. BMC Med Res Methodol. 2020 Jul 2;20(1):177. doi: 10.1186/s12874-020-01059-y. PMID: 32615936 [This article was not found in the References Section]

Abstract Section:  

The statement “This paper analysis, from the bibliometric viewpoint, those publications that relate COVID-19 and eye indexed in Scopus since the pandemic began and identify the resulting main research lines” lacks a verb and so is not a complete sentence.  The statement “growth is expected to increase”, also seen in the Conclusions Section, was not supported by the appearance of words “growth” or “increase” or “extrapolation” or graph of items versus time, in the Results Section or in the Discussion Section.  

It is expected that full original articles and comprehensive reviews will have as many as 200 references, and that other items such as brief reports, case reports, and editorials will have as few as 5 references. Calculating a total mean or average of 8 references among all such items appears of limited use.   

Introduction Section: 

The Introduction Section should include some knowledge from the 4 articles listed above. 

It may be better to present what has been previously published on covid bibliometrics and any unanswered questions about bibliometrics and utility of bibliometric information. The introduction Section does not need a day-by-day recounting of the emergence, description, characterization, and progression from cases, to cluster, outbreak, epidemic, pandemic, then endemic.  

Methods Section:  

A complete listing of all keywords used and any keywords not used would be appropriate.  

Line 81 “records were exported as csv to then be imported to the VOSviewer networks visualisation programme”. Abbreviations should be defined at first use, such as “csv” (comma-separated values file) and “h index” [“A scientist has index h if h of his or her NP (number of papers) papers have at least h citations each and the other (NP – h) papers have fewer than h citations each. E.g. a h-index of 20 means the researcher has 20 papers each of which has been cited 20+ times.” https://libguides.sun.ac.za/c.php?g=742955&p=5316861] [“defined as the highest number of publications of a scientist that received h or more citations each while the other publications have not more than h citations each." For example, a scholar with an h-index of 5 had published 5 papers, each of which has been cited by others at least 5 times.” “h-Index - Bibliometrics and Altmetrics: Measuring the Impact…” July 8 2022 https://lib.guides.umd.edu/bibliometrics/h-index#:~:text=%22It%20is%20defined%20as%20the,others%20at%20least%205%20times.]  

Results Section:  

There is no need to redundantly restate in text the same numbers presented in tables or figures.  

Figures 1, 2, there is no need to present pie charts or bar charts or maps the numbers or percentages of published articles or keyword topics that can be more legibly listed in Tables.   

Figures 3, 4, 6, 7, 9, showing “network visualization” have so many and lines or dots and words, some of which are small, as to be difficult to decipher.  

Tables 1, 2, 3, 4, 5, alignment of words to the left, left justification, would be more legible; alignment of numbers to the right, right justification, would be more legible.  

Table 3, column headings appear out of alignment with the content.   

Table 3, has a blank space appearing as a large shaded box and some displaced text “NCit: Citations received about the search results”.   

Table 5, the boxes that are red or green or blue are distracting. Traditional groupings of topics are: etiology and causative organism; epidemiology; symptoms and signs; clinical pathology, gross pathology and microscopic pathology; diagnosis and prognosis; treatment and prevention.   

Discussion Section: 

The claim that “This study is the first bibliometric analysis of the world’s scientific production on COVID-19 and the ocular / visual aspects and manifestations indexed in Scopus” may not be true, because there is a previously published article on covid ophthalmology and bibliometrics: Ali Forouhari, Vahid Mansouri, Sare Safi, Hamid Ahmadieh, Amir Ghaffari Jolfayi. A Systematic Literature Review and Bibliometric Analysis of Ophthalmology and COVID-19 Research. Journal of Ophthalmology. 2022 May 24; 2022: 8195228. doi: 10.1155/2022/8195228. eCollection 2022. 

Recommend to summarize results of previously published articles on covid bibliometrics and compare and contrast for similarities and differences in this present manuscript.  

The Discussion Section should compare and contrast for similarities and differences between this present manuscript and the 4 articles listed above. 

Language usage in general: 

Needs editing to improve word choices, from vernacular, such as “of them all” “most productive journal” “obtained almost 20% of citations”.  

 Eliminate adjectives and superlative such as “thorough” “plenty” “first…of the world’s”.  

 Eliminate any word that is not essential to convey meaning, such as “viewpoint” “mainly”. 

 Eliminate any phrase that is not essential or not specific or could be misunderstood, such as “main research lines” “using only documents published in any language” “largest publication output”. 

Subsection 2.1 Heading is set in italic roman type. Subsection 2.2 Heading is set in regular roman type. Consistency in typesetting is needed.

The first appearance of SCOPUS is in uppercase letters.  The next uses of Scopus is lowercase.  Need consistent use of type setting.

Line-Specific Comments:

Line 14.  Rather than “contributed” consider the word “originated”

Lines 30-33 “Since then and to the present-day (28 december 2021), there have been more than 281 million contagions and 5.4 million people have died from this disease in the world[1], athough the WHO considers that the number of people who have died directly or indirectly from this disease would be triple the official figure.”  The “present-day” of “28 december 2021” is no longer current.  Please revise the sentence to be more current.

Lines 34-35 “As the virus rapidly spread around the world, many researchers focused their activ-ity on knowing this disease in an attempt to control the pandemic.” Consider replacement of the word “knowing” with the word “understanding”.

Line 42 “SCOPUS”.  Line 59 “Scopus”.  Line 66 “Scopus”.

Line 73 ““CoronaVIrus”.  Typographic error?

Line 227 “Table 3. Top 10 cited articles in the eye-related COVID-19 literature (2019-2021).”  Please specify in this table title the calendar date or data accession date on which it was decided which publications are the 10 most cited publications.  The study period was stated to be “1 January 2020 to 16 March 2021”.  In the Introduction, line 30, is a calendar date of 28 december 2021. This manuscript draft is being reviewed in July of 2022.   At line 265 “The search was repeated in October 2021 and the monthly rate of articles rose in 2021 from 185 to 266.”

Line 228 ” Table 4. Top 10 most relevant articles (Scopus) in the eye-related COVID-19 literature (2019-2021).”  Please specify in this table title the calendar date or data accession date on which it was decided which publications are the 10 most relevant publications. The study period was stated to be “1 January 2020 to 16 March 2021”.  This manuscript draft is being reviewed in July of 2022

Line 228, Table 4.  The table content is a mix of lower case and upper case lettering

Line 228, Table 4.  Please include a definition of the abbreviation “NCit” in the table title or in a footnote

Line 228 “NCit” Line 337 “Ncit”

Line 229-231 “3.5. Analysis of Co-ocurrence of Terms” “Figure 8 shows a co-occurence map where the chosen analysis unit was authors’ Key-words. The term COVID-19 appears as the most important node with 890 ocurrences.”  Spelling: “ocurrence” “occurrence”

Figure 8. Is this a typographic error? “Personal Protective Equipr”

Lines 270-272 “India normally occupies a lower position in this ranking (7th in the set of all disciplines and 11th on Medicine). Collaboration among countries is headed by the USA, and the most well-eatablished co-authorship is that of the USA and the UK.”  Please change the poor word choice “normally”.  Please correct the spelling error.

Line 291-292 “The top 10 most cited publications contain some documents in which the eye subject was not the main subject.  Please specify the calendar date on which it was decided which publications are the 10 most cited publications.

Lines 332-335 “Another of its limitations is that assigning the country to which each document is attributed is done with author affiliations, which favours international collaborations because one same work is assigned to different countries, unlike publications by teams with the same affiliation[54].” Please revise this sentence to improve the clarity of writing.

Line 351 “and the effect of vaccination on it become known.”  Delete the phrase “on it”

Reviewer 3 Report

Thank you so much for giving me the opportunity to review this paper. In this manuscript, the authors presented a well described visual and bibliometric analysis of current literature evidence on the relationship between COVID-19 and eye. 

The manuscript is well written and I have only some minor concerns to be addressed before publication:

1) Methods: improve description of visualization map (how was centrality defined?)

2) Results: add a brief paragraph describing the main characteristics of studies reported in table 3 (those with the highest number of citations).

3) Discussion: Which are the 2-3 main messages emerging from current research and which is the direction traced for future studies?

4) Minor changes:

a.      Line 25: change “been” with were.

b.      Line 43: add reference if this comes from a previous study; instead, if your group performed the bibliographic search, it is not the place to show that, and simply remove lines 41-43.

c.      Line 50: write out the acronym OCT.

d.      Line 50: change “line” with “research line”.

e.      Check the grey box in the last row of table 3

Round 2

Reviewer 1 Report

The authors have sufficiently taken into account the comments of the reviewers, although somewhat more could have be done. Nevertheless I think the manuscript is now suitable for publication. 

Author Response

We would like to acknowledge the reviewer for the positive comments and the final recommendation of our manuscript.

Reviewer 2 Report

The authors have improved the writing.  A professional copy writer could improve word choices.

Author Response

We acknowledge the reviewer’s comments. The manuscript has been revised again by a native English speaker and some typos has been corrected. We would also thank her final recommendation of the manuscript.
